# LEVERAGING CLASS HIERARCHIES
# WITH METRIC-GUIDED PROTOTYPE LEARNING

## ABSTRACT

In many classification tasks, the set of classes can be organized according to a meaningful hierarchy. This structure can be used to assess the severity of confusing each pair of classes, and summarized under the form of a *cost matrix* which also defines a finite metric. We propose to integrate this metric in the supervision of a *prototypical network* in order to model the hierarchical class structure. Our method relies on jointly learning a feature-extracting network and a set of class representations, or *prototypes*, which incorporate the error metric into their relative arrangement in the embedding space. We show that this simultaneous training allows for consistent improvement of the severity of the network's errors with regard to the class hierarchy when compared to traditional methods and other prototype-based strategies. Furthermore, when the induced metric contains insight on the data structure, our approach improves the overall precision as well. Experiments on four different public datasets—from agricultural time series classification to depth image semantic segmentation—validate our approach.

## 1   INTRODUCTION

Most classification models focus on maximizing the prediction accuracy, regardless of the semantic nature of errors. This can lead to high performing models, but puzzling errors such as confusing a tiger and a sofa. This casts doubt on what a model actually actually understands from the required task and data distribution. Neural networks in particular have been criticized for their tendency to produce improbable yet confident errors, notably when attacked (Akhtar & Mian, 2018).

The classes of most classification problems can be organized according to a hierarchical structure. Such tree-shaped taxonomy of concepts can be generated by domain experts, or automatically from class names with the WordNet graph (Miller et al., 1990) or word embeddings (Mikolov et al., 2013). A step towards more reliable and interpretable algorithms would be to explicitly model the difference of gravity between errors, as defined by a hierarchical nomenclature.

For a classification task over a set $\mathcal{K}$ of $K$ classes, the hierarchy of errors can be encapsulated by a cost matrix $D \in \mathbb{R}_+^{K \times K}$, defined such that the cost of predicting class $k$ when the true class is $l$ is $D[k, l] \geq 0$, and $D[k, k] = 0$ for all $k = 1 \cdots K$. Among many other options (Kosmopoulos et al., 2015), one can define $D[k, l]$ as the length of the shortest path between the nodes corresponding to classes $k$ and $l$.

As pointed out by Bertinetto et al. (2020), the first step towards algorithms aware of hierarchical structures would be to generalize the use of cost-based metrics. For example, early iterations of the ImageNet challenge (Russakovsky et al., 2015; Deng et al., 2010) proposed to weight errors according to hierarchy-based costs. For a dataset indexed by $\mathcal{N}$, the *Average Hierarchical Cost* (AHC) between class predictions $y \in \mathcal{K}^{\mathcal{N}}$ and the true labels $z \in \mathcal{K}^{\mathcal{N}}$ is defined as:

$$\text{AHC}(y, z) = \frac{1}{|\mathcal{N}|} \sum_{n \in \mathcal{N}} D[y_n, z_n] \,. \tag{1}$$

Along with evaluation metrics, loss functions should also take the cost matrix into account. While it is common to focus on retrieving certain classes through weighting schemes, preventing specific class confusions is less straightforward. The cross entropy with one-hot target vectors for example singles out the prediction with respect to the correct class, but treats all other classes equally. Beyond reducing the AHC, another advantage of incorporating the class hierarchy into the learning phase

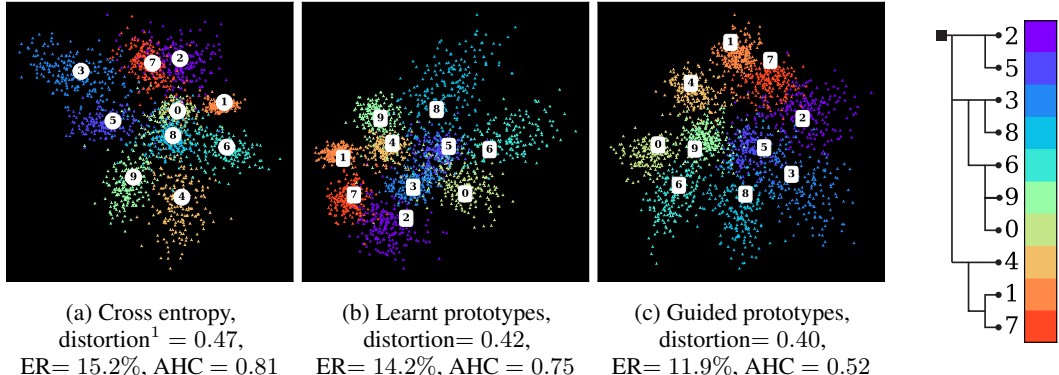

| (a) Cross entropy, | (b) Learnt prototypes, | (c) Guided prototypes, |
|---|---|---|
| distortion[1] $= 0.47$, | distortion$= 0.42$, | distortion$= 0.40$, |
| ER$= 15.2\%$, AHC $= 0.81$ | ER$= 14.2\%$, AHC $= 0.75$ | ER$= 11.9\%$, AHC $= 0.52$ |

Figure 1: Mean class representation ○ , prototypes □ , and 2-dimensional embeddings ⋏ learnt on perturbed MNIST by a 3-layer convolutional net with three different classification modules: (a) cross-entropy, (b) learnt prototypes, and (c) learnt prototypes guided by a tree-shaped taxonomy (constructed according to the authors' perceived visual similarity between digits). The guided prototypes (c) embed more faithfully the class hierarchy: classes with low error cost are closer[1]. This is associated with a decrease in the *Average Hierarchical Cost* (AHC), as well as *Error Rate* (ER), indicating that our taxonomy may contain useful information for learning better visual features.

is that $D$ may contain information about the structure of the data as well. Though it is not always the case, co-hyponyms (i.e. siblings) in a class hierarchy tend to share some structural properties. Encouraging such classes to have similar representations could lead to more efficient learning, *e.g.* by pooling common feature detectors. Such priors on the class structure may be especially crucial when dealing with a large taxonomy, as noted by Deng et al. (2010).

In this paper, we introduce a method to integrate the class hierarchy into a classification algorithm. We propose a new scale-free, distortion-based regularizer for prototypical network (Yang et al., 2018; Chen et al., 2019). This penalty allows the network to learn prototypes organized such that their relative distances reflect their distance in a class hierarchy.

The contributions of this paper are as follows:

- We introduce a scale-independent formulation of the distortion between two metric spaces, and an associated smooth regularizer.
- This formulation allows us to incorporate knowledge of the class hierarchy into a neural network at no extra cost in trainable parameters and computation.
- We show on four public datasets (CIFAR100 , NYUDv2, S2-Agri, and iNaturalist-19) that our approach decreases the average cost of the prediction of standard backbones.
- As illustrated in Figure 1, we show that our approach can also lead to a better (unweighted) precision, which we attribute to useful priors contained in the taxonomy of classes.

## 2 RELATED WORK

**Prototypical Networks:** Our approach builds on the growing corpus of work on prototypical networks. These models are deep learning analogues of nearest centroid classifiers (Tibshirani et al., 2002) and Learning Vector Quantization networks (Sato & Yamada, 1995; Kohonen, 1995), which associate to each class a representation, or prototype, and classify observations according to the nearest prototype. These networks have been successfully used for few-shot learning (Snell et al., 2017; Dong & Xing, 2018), zero-shot learning (Jetley et al., 2015), and supervised classification (Guerriero et al., 2018; Yang et al., 2018; Mettes et al., 2019; Chen et al., 2019).

In most approaches, the prototypes are directly defined as the centroid of the learnt representations of samples of their classes, and updated at each episode (Snell et al., 2017) or iteration (Guerriero et al.,

---

[1]For a formal definition of our scale-free distortion, see Section 3.2; computed from the means of class embeddings for the cross entropy.

2018). In the work of Mettes et al. (2019) and Jetley et al. (2015), the prototypes are defined prior to learning the embedding function. In this work, we follow the approach of Yang et al. (2018) and learn the prototypes simultaneously to the data embedding function.

**Hierarchical Priors:**   The idea of exploiting the latent taxonomic structure of semantic classes in order to improve the accuracy of a model has been extensively explored (Silla & Freitas, 2011), from traditional Bayesian modeling (Gelman et al., 2013, Chapter 5) to adaptive deep learning architectures (Yan et al., 2015; Roy et al., 2020; Salakhutdinov et al., 2012; Ayub & Saini, 2011). However, for these neural networks, the hierarchy is discovered by the network itself in the goal of improving the overall accuracy of the model. In our setting, the hierarchy is defined a priori, and serves both to evaluate the quality of the model and to guide the learning process towards a reduced prediction cost.

Srivastava & Salakhutdinov (2013) propose implementing Gaussian priors on the weight of neurons according to a fixed hierarchy. Redmon & Farhadi (2017) implement an inference scheme based on a tree-shaped graphical model derived from a class taxonomy. Closest to our work, Hou et al. (2016) propose a regularization based on the earth mover distance to penalize errors with high costs.

More recently, Bertinetto et al. (2020) highlighted the relative lack of well-suited methods for dealing with hierarchical nomenclatures in the deep learning literature. They advocate for a more widespread use of the AHC for evaluating models, and detail two simple baseline classification modules able to decrease the AHC of deep models: *Soft-Labels* and *Hierarchical Cross-Entropy*. See Section 4.3 for more details on these schemes.

**Hyperbolic Prototypes:**   Motivated by the quality of their low-dimensional embedding of hierarchical data structures (De Sa et al., 2018), hyperbolic spaces are at the center of recent advances in modeling hierarchical relations (Nickel & Kiela, 2017; Khrulkov et al., 2020). Closer to this work, (Liu et al., 2020; Long et al., 2020) also propose to embed a class hierarchy into the latent representation space. However, both approaches embed the class hierarchy before training the data embedding network. In contrast, we argue that incorporating the hierarchical structure during the training of the model allows the network and class embeddings to share their respective insights, leading to a better trade-off between AHC and accuracy. In this paper, we only explore this claim in Euclidean geometry, as this setting allows for the seamless integration of our method.

**Finite Metric Embeddings:**   Our objective of computing class representations with pairwise distances determined by a cost matrix has links with finding an isometric embedding of the cost matrix—seen as a finite metric. This problem has been extensively studied (Indyk et al., 2017; Bourgain, 1985) and is at the center of the growing interest for hyperbolic geometry (De Sa et al., 2018). However, our goal is simply to influence the learning of prototypes with a metric rather than necessarily seeking the best possible isometry.

## 3   METHOD

We consider a generic dataset $\mathcal{N}$ of $N$ elements $x \in \mathcal{X}^{\mathcal{N}}$ with ground truth classes $z \in \mathcal{K}^{\mathcal{N}}$. The classes $\mathcal{K}$ are organized along a tree-shape hierarchical structure, allowing us to define a cost matrix $D$ by considering the shortest path between nodes. The matrix thus defined is symmetric, with a zero diagonal and strictly positive elsewhere, and also respects the triangle inequality: $D[k,l] + D[l,m] \geq D[k,m]$ for all $k, l, m$ in $\mathcal{K}$. In other words, $D$ defines a finite metric. We denote by $\Omega$ the *embedding space* which, when equipped with the distance function $d : \Omega \times \Omega \mapsto \mathbb{R}_+$, forms a metric space as well.

### 3.1   PROTOTYPICAL NETWORKS

A prototypical network is characterized by an embedding function $f : \mathcal{X} \mapsto \Omega$, typically a neural network, and a set $\pi \in \Omega^{\mathcal{K}}$ of $K$ prototypes. $\pi$ must be chosen such that any sample $x_n$ of a given class $k$ has a representation $f(x_n)$ which is *close* to $\pi_k$ and *far* from other prototypes.

Following the methodology of Snell et al. (2017), a prototypical network $(f, \pi)$ associates to an observation $x_n$ the following posterior distribution over its class $z_n$:

$$p(z_n = k | x_n) = \frac{\exp\left(-d\left(f(x_n), \pi_k\right)\right)}{\sum_{l \in \mathcal{K}} \exp\left(-d\left(f(x_n), \pi_l\right)\right)}, \forall k \in \mathcal{K}. \tag{2}$$

We can then define an associated loss with the normalized negative log-likelihood of the true classes:

$$\mathcal{L}_{\text{data}}(f, \pi) = \frac{1}{N} \sum_{n \in \mathcal{N}} \left( d(f(x_n), \pi_{z_n}) + \log\left(\sum_{l \in \mathcal{K}} \exp\left(-d(f(x_n), \pi_l)\right)\right) \right). \tag{3}$$

This loss is such that the representation $f(x_n)$ is attracted towards the prototype of the class $z_n$, while it is repelled by the other prototypes. Conversely, prototype $\pi_k$ is drawn towards the representations $f(x_n)$ of samples $n$ of class $k$ and away from the representations of other classes.

Following the insights of Yang et al. (2018), the embedding function $f$ and the prototypes $\pi$ are learned *simultaneously*. This differs with many works on prototypical networks which learn prototypes separately or define them as centroids of representations. We take advantage of this joint training to learn prototypes which take into account both the distribution of the data and the relationships between classes, as described in the next section.

## 3.2 METRIC-GUIDED PENALIZATION

We propose to incorporate the cost matrix $D$ into a regularization term in order to encourage the prototypes to organize in the embedding space $\Omega$ in a manner that is consistent with the finite metric defined by $D$. Since sample representations are attracted to their respective prototypes in (3), this regularization also affects the embedding network.

**Scale-Free Distortion** As described in De Sa et al. (2018), the distortion of a mapping $k \mapsto \pi_k$ between the finite metric space $(\mathcal{K}, D)$ and the continuous metric space $(\Omega, d)$ can be defined as:

$$\text{disto}(\pi, D) = \frac{1}{K(K-1)} \sum_{k,l \in \mathcal{K}^2, \, k \neq l} \frac{|d(\pi_k, \pi_l) - D[k,l]|}{D[k,l]}. \tag{4}$$

We argue that prototypes arrangements $\pi$ with low distortion incur lower hierarchical costs. Let us first consider a misclassified sample $x_n$ of true class $k$. Since $x_n$ is misclassified, its representation $f(x_n)$ is closer to another prototype than to the true prototype $\pi_k$. However, we can assume that $f(x_n)$ is still closer to $\pi_k$ than to *most* prototypes, as the loss $\mathcal{L}_{\text{data}}$ would starkly penalize $f$ otherwise. Since low distortion prototypes group together prototypes of classes with small mutual error cost, the erroneous class predicted will likely be of low hierarchical cost with respect to $k$.

However, achieving low-distortion also imposes a specific scale on the distances between prototypes in the embedding space. This scale may conflict with the second term of $\mathcal{L}_{\text{data}}$ which encourages distances between embeddings and unrelated prototypes to be as large as possible. Therefore, lower distortion may also cause lower precision. To remove this imposed scaling, we introduce a scale-independent formulation of the distortion:

$$\text{disto}^{\text{scale-free}}(\pi, D) = \min_{s \in \mathbb{R}_+} \text{disto}(s \cdot \pi, D), \tag{5}$$

where $s \cdot \pi$ are the scaled prototypes, whose coordinates in $\Omega$ are multiplied by a scalar factor $s$. As shown in the appendix, $\text{disto}^{\text{scale-free}}$ can be efficiently computed algorithmically.

**Distortion-Based Penalization** We propose to incorporate the error qualification $D$ into the prototypes' relative arrangement by encouraging a low *scale-free* distortion between $\pi$ and $D$. To this end, we define the following smooth surrogate of $\text{disto}^{\text{scale-free}}$:

$$\mathcal{L}_{\text{disto}}(\pi) = \frac{1}{K(K-1)} \min_{s \in \mathbb{R}_+} \sum_{k,l \in \mathcal{K}^2, \, k \neq l} \left( \frac{sd(\pi_k, \pi_l) - D[k,l]}{D[k,l]} \right)^2. \tag{6}$$

$\mathcal{L}_{\text{disto}}$ can be computed in closed form as a function of $\pi$ and can thus be directly used as regularizer.

### 3.3 END-TO-END TRAINING

We combine $\mathcal{L}_{\text{data}}$ and $\mathcal{L}_{\text{disto}}$ in a single loss $\mathcal{L}$. $\mathcal{L}_{\text{data}}$ allows to jointly learn the embedding function $f$ and the class prototypes $\pi$, while $\mathcal{L}_{\text{disto}}$ enforces a metric-consistent prototype arrangement:

$$\mathcal{L}(f, \pi) = \mathcal{L}_{\text{data}}(f, \pi) + \lambda \mathcal{L}_{\text{disto}}(\pi) , \tag{7}$$

with $\lambda \in \mathbb{R}_+$ an hyper-parameter setting the strength of the regularization.

### 3.4 CHOOSING A METRIC SPACE

Prototypical networks operating on $\Omega = \mathbb{R}^m$ typically use the squared Euclidean norm in the distance function, motivated by its quality as a Bregman divergence (Snell et al., 2017). However, we observe that defining $d$ with the Euclidean norm yields significantly better results. The non-differentiability can be handled by composing with a Huber-like (Huber et al., 1973; Charbonnier et al., 1997) function $d = H(\|\cdot\|)$, with $H$ defined as:

$$H(x) = \delta(\sqrt{\|x\|^2/\delta^2 + 1} - 1) , \tag{8}$$

and $\delta \in \mathbb{R}_+$ a (small) hyper-parameter. The resulting metric $d$ is asymptotically equivalent to the Euclidean norm for large distances, and behaves like the smooth squared Euclidean norm for small distances. In Section 4.5, we investigate the effect of this change.

## 4 EXPERIMENTS

### 4.1 DATASETS AND BACKBONES

We evaluate our approach with different tasks on public datasets with fine-grained class hierarchies: image classification on CIFAR100 (Krizhevsky et al., 2009) and iNaturalist-19 (Van Horn et al., 2018), RGB-D image segmentation on NYUDv2 (Nathan Silberman & Fergus, 2012), and image sequence classification on S2-Agri (Sainte Fare Garnot et al., 2020). We define the cost matrix of these class sets as the length of the shortest path between nodes in the tree-shape taxonomies, represented in the Appendix. As shown in Table 1, these datasets cover different settings in terms of data characteristics, as well as tree structures.

**Illustrative Example on MNIST:**  In Figure 1 and Figure **??** of the Appendix, we illustrate the difference in performance and embedding organization for different approaches. We use a small 3-layer convolutional net trained on MNIST with random rotations (up to 40 degrees) and affine transformations (up to 1.3 scaling). For plotting convenience, we set the feature's dimension to 2.

**Image Classification on CIFAR100:**  CIFAR100 is composed of 50 000 training images and 10 000 test images of size $32 \times 32$, evenly distributed across 100 classes. We use a super-class system inspired by Krizhevsky et al. (2009) and form a 5-level hierarchical nomenclature of size: 2, 4, 8, 20, and 100 classes. We use as backbone the established ResNet-18 (He et al., 2016) for this dataset.

**Semantic Segmentation on NYUDv2:**  NYUDv2 is an RGB-D image segmentation dataset composed of 1 449 pairs of RGB images of indoor scenes and their corresponding depth maps. We use the standard split of 795 training and 654 testing pairs. We combine the 4 and 40 class nomenclatures of Gupta et al. (2013) and the 13 class system defined by Handa et al. (2016) to construct a 3-level hierarchy. We use FuseNet (Hazirbas et al., 2016) as backbone for this dataset.

**Image Sequence Classification on S2-Agri:**  S2-Agri comprises 189 971 sequences of multi-spectral satellite images of agricultural parcels. We define a 4-level hierarchy of size 4, 12, 19, and 44 classes to organize the crop types. This class hierarchy is critical to monitoring agencies, as it is related to the level of subsidy allocated to farmers. We use the PSE+TAE architecture (Sainte Fare Garnot et al., 2020) as the backbone, and follow their 5-fold cross-validation scheme for training.

**Fine-Grained Image Classification on iNaturalist-19 (iNat-19)**  iNat-19 (Van Horn et al., 2018) is a fine-grained image classification dataset comprised of 265 213 images of living organisms. The images are labeled by experts and users of the *iNaturalist* app, using a fine-grained taxonomy. iNat-19

contains 1 010 different classes, organized into a hierarchy of 7 levels with respective width 3, 4, 9, 34, 57, 72, and 1 010. We use ResNet-18 pre-trained on ImageNet as backbone. We sample 75% of available images for training, while the rest is evenly split into a validation and test set.

Table 1: Data composition and nomenclature of the four studied datasets. IR stands for the Imbalance Ratio (largest over smallest class count), nodes and leaves denote respectively the total number of classes and leaf-classes in the tree-shape hierarchy, ABF stands for the Average Branching Factor, and $\langle D \rangle$ stands for the average pairwise distance.

| Dataset | Data | | | Hierarchical Tree | | |
| | Volume (Gb) | IR | Depth | Nodes (leaves) | ABF | $\langle D \rangle$ |
|---|---|---|---|---|---|---|
| NYUDv2 | 2.8 | 93 | 3 | 57 (40) | 5.0 | 4.3 |
| S2-Agri | 28.2 | 617 | 4 | 83 (45) | 5.8 | 6.5 |
| CIFAR100 | 0.2 | 1 | 5 | 134 (100) | 3.8 | 7.0 |
| iNat-19 | 82.0 | 31 | 7 | 1189 (1010) | 6.6 | 11.0 |

## 4.2 HYPER-PARAMETERIZATION

The embedding space $\Omega$ is chosen as $\mathbb{R}^{512}$ for iNat-19 and $\mathbb{R}^{64}$ for all other datasets. We define $d$ as the Euclidean norm (see 4.5 for a discussion on this choice). We evaluate our approach (Guided-proto) with $\lambda = 1$ in (7) for all datasets. We use the same training schedules and learning rates as the backbone networks in their respective papers. In particular, the class imbalance of S2-Agri is handled with a focal loss (Lin et al., 2017).

## 4.3 COMPETING METHODS

In the paper where they are introduced, all backbone networks presented in Section 4.1 use a linear mapping between the samples embedding and the class scores, as well as the cross-entropy loss. The resulting performance of these networks serves as baseline to estimate the gains in Average Hierarchical Cost (AHC) and Error Rate (ER) provided by our approach and other competing methods we re-implemented.

- **Hierarchical Cross-Entropy** (HXE) Bertinetto et al. (2020) model the class structure with a hierarchical loss composed of the sum of the cross-entropies at each level of the class hierarchy. As suggested, a parameter $\alpha$ taken as $0.1$ defines exponentially decaying weights for higher levels.

- **Soft Labels** (Soft-labels) Bertinetto et al. (2020) propose as second baseline in which the the one-hot target vectors are replaced by soft target vectors in the cross-entropy loss. These target vectors are defined as the softmin of the costs between all labels and the true label, with a temperature $1/\beta$ chosen as $0.1$, as recommended in Bertinetto et al. (2020).

- **Earth Mover Distance regularization** (XE+EMD): Hou et al. (2016) propose to account for the relationships between classes with a regularization based on the squared earth mover distance. We use $D$ as the ground distance matrix between the probabilistic prediction $p$ and the true class $y$:

$$\mathcal{L}_{EMD}(p, y) = \frac{1}{N} \sum_{n \in \mathcal{N}} \sum_{k=1}^{K} p(z_n = k | x_n)^2 (D[k, y] - \mu) .$$

  This regularizer is added along the cross-entropy with a weight of $0.5$ and an offset $\mu$ of $3$.

- **Hierarchical Inference** (YOLO): Redmon & Farhadi (2017) propose to model the hierarchical structure between classes into a tree-shaped graphical model. First, the conditional probability that a sample belongs to a class given its parent class is obtained with a softmax restricted to the class' co-hyponyms (*i.e.* siblings). Then, the posterior probability of a leaf class is given by the product of the conditional probability of its ancestors. The loss is defined as the cross-entropy of the resulting probability of the leaf-classes.

- **Hyperspherical Prototypes** (Hyperspherical-proto): The method proposed by Mettes et al. (2019) is closer to ours, as it relies on embedding class prototypes. They advocate to first position prototypes on the hypersphere using a rank-based loss (see Section 4.5) combined with a prototype-separating term. They then use the squared cosine distance between the image embeddings and

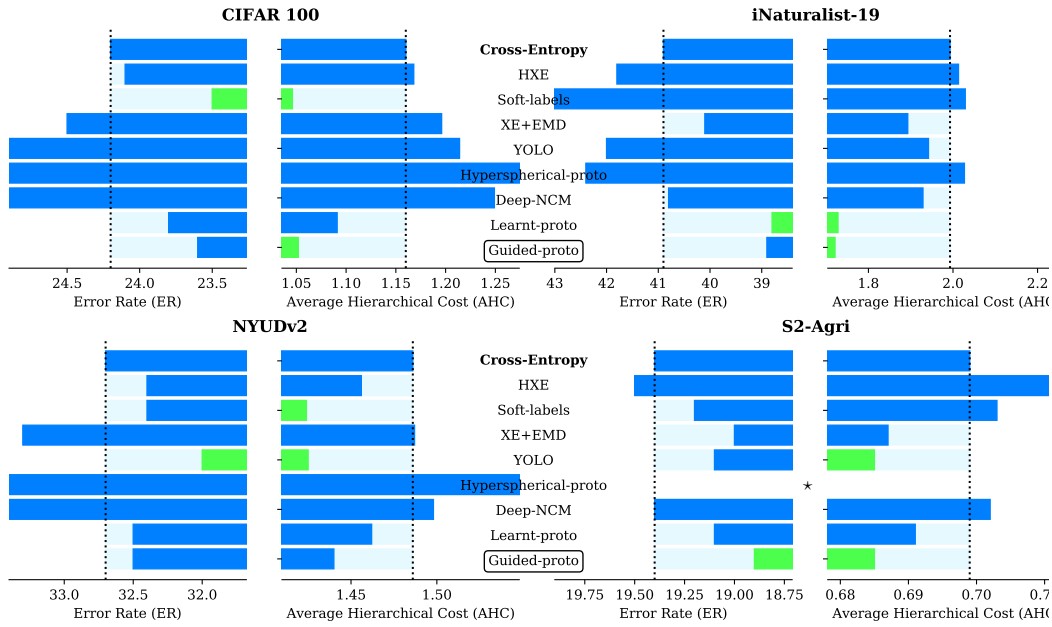

Figure 2: Error Rate (ER) in % and Average Hierarchical Cost (AHC) on four datasets for our proposed method (framed), the Cross-Entropy baseline (in bold), and the competing approaches. The best performances on each dataset are plotted in green. Our guided prototype approach improves both the ER and AHC across the four datasets compared to the baseline. The metrics are computed with the median over 5 runs for CIFAR100, the average over 5 cross-validation folds for S2-Agri, and a single run for NYUDv2 and iNat-19. The numeric values are given in the Appendix. ($\star$ not evaluated on S2-Agri).

prototypes to train the embedding network. Note that in our re-implementation, we used the finite metric defined by $D$ instead of Word2Vec (Mikolov et al., 2013) embeddings to position prototypes. Lastly, we do not evaluate on S2-Agri as the integration of the focal loss is non-trivial.

- **Deep Mean Classifiers** (Deep-NCM): Guerriero et al. (2018) present another prototype-based approach. Here, the prototypes are the cumulative mean of the embeddings of the classes' samples, updated at each iteration. The embedding network is supervised with $\mathcal{L}_{data}$ with $d$ defined as the squared Euclidean norm.

- **Prototype Learning**(Learnt-proto): Learnt prototypes without regularization, *i.e.* $\lambda = 0$ in (7).

### 4.4 ANALYSIS

**Overall Performance:** As displayed in Figure 2, the benefits provided by our approach can be appreciated on all datasets. Compared to cross-entropy, our metric-guided prototype models improve the AHC by $3\%$ on NYUDv2 and S2-Agri, and up to $9\%$ and $14\%$ for CIFAR100, and iNat-19 respectively. The hierarchical inference scheme of Redmon & Farhadi (2017) performs on par or better than our methods for NYUDv2 and S2-Agri, while Soft-labels perform well on CIFAR100 and NYUDv2. Yet, only the metric guided prototype approach brings a consistent reduction of the hierarchical cost across the four datasets. This suggest that arranging the embedding space consistently with the cost metric is a robust way of reducing a model's hierarchical error cost.

While being initially designed to reduce the AHC, our methods also improve the ER by 3 to $4\%$ across all datasets compared to the cross-entropy baseline. This indicates that cost matrices derived from the class hierarchy can indeed help neural networks to learn better representations.

**Prototype Learning:** We observe that learnt prototype approach Learnt-proto consitently outperforms the Deep-NCM method. This suggest that defining prototypes as the centroids of their class representations might actually be disadvantageous. As illustrated on Figure 1c, the positions of the

Table 2: Influence of the choice of scaling in $\mathcal{L}_{\text{disto}}$, metric guiding regularizer, and distance function $d$ on the performance of Guided-proto on the four datasets. For $d$, we compare the performance of the Euclidean norm, the pseudo-Huberized Euclidean norm, and the square Euclidean norm.

|  | CIFAR100 | | NYUDv2 | | S2-Agri | | iNat-19 | |
|---|---|---|---|---|---|---|---|---|
|  | ER | AHC | ER | AHC | ER | AHC | ER | AHC |
| **Guided-proto** | 23.6 | **1.052** | 32.5 | 1.440 | **18.9** | **0.685** | **38.9** | 1.721 |
| Fixed-scale | +0.1 | +0.003 | 0.0 | 0.000 | +0.2 | **+0.001** | +0.9 | 0.000 |
| Rank-based guiding | **-0.3** | +0.004 | +0.2 | +0.005 | +0.2 | +0.006 | +0.4 | **-0.003** |
| Fixed-proto | +1.1 | +0.031 | +0.6 | +0.013 | +0.5 | +0.025 | +5.0 | +0.427 |
| Pseudo-Huber | +0.1 | +0.015 | **-0.3** | **-0.017** | +0.4 | +0.016 | +0.2 | +0.003 |
| Squared Norm | +1.0 | +0.118 | 0.0 | +0.005 | +0.6 | +0.022 | +2.2 | +0.233 |

embeddings follow a Voronoi partition (Fortune, 1992) with respect to the learnt prototypes rather than the prototypes being at the centroid of representations. A surprising observation for us is that Learnt-proto consistently outperforms the cross-entropy, both in terms of AHC and ER.

**Computational Efficiency:** Computing distances to prototypes is comparable in terms of computation time to computing a linear mapping. Consequently, both training and inference time are equivalent when using the cross-entropy or guided-proto-disto, which is 2% faster.

### 4.5 ABLATION STUDY

**Fixed-Scale Distortion:** In Table 2, we compare the performance of our *scale-free* regularizer to an alternative version of $\mathcal{L}_{\text{disto}}$ in which the scale remains fixed to $s = 1$. Across datasets, this results in an increased error rate, which we attribute to prototype distances being fixed by $\mathcal{L}_{\text{disto}}$. The benefit of our scale-free regularizer is especially valuable for the complex class structure of iNat-19, improving the overall classification accuracy by 1 point compared to the fixed-scale version.

**Rank-based Regularization:** Mettes et al. (2019) use a rank-based loss (Burges et al., 2005) to encourage prototype mappings whose pairwise distance follows the same order as an external qualification of errors $D$. We argue that our formulation of $\mathcal{L}_{\text{disto}}$ provides a stronger supervision than only considering the order of distances, and allows the prototypes to find a more profitable arrangement in the embedding space. In Table 2, we observe that replacing our distortion-based loss by a rank-based one results in a slight decrease of overall performance.

**Guided vs. fixed prototypes :** Our experiments confirm the benefit of jointly learning the prototypes and the embedding network instead of learning the prototypes first as Hyperspherical-proto. We also evaluate the performance of Fixed-proto in Table 2, for which we first fix the prototypes with $\mathcal{L}_{\text{disto}}$ and then train the network with $\mathcal{L}_{\text{data}}$. While this approach reduces the AHC compared to the cross-entropy baseline in some cases, its ER is consistently higher than for guided methods. This suggests that insights from the data distribution can conversely benefit the positioning of prototypes, and that they should be learned conjointly.

**Choice of distance :** In Table 2, we report the performance of the Guided-proto model on the four datasets when replacing the Euclidean norm alternately with the squared Euclidean norm and an Huberized Euclidean distance (with $\delta = 0.1$ in (8)). Across our experiments, the squared-norm based model yields a worse performance. This is a notable result as it is the distance commonly used in most prototypical networks (Snell et al., 2017; Guerriero et al., 2018). The Huberized norm performs worse than the Euclidean distance on all dataset, with the exception NYUDv2.

**Further Ablation Study:** In the Appendix we present an extended ablation study, notably showcasing the resilience of our approach to a wide-range of hyper-parameters value.

## 5 CONCLUSION

We introduced a new regularizer to incorporate the class hierarchy during the training of a prototypical network. We showed that our methods consistently decreased the average hierarchical cost of three different backbone networks on different tasks and four datasets. Furthermore, our approach can reduce the rate of errors as well. In contrast to most recent literature on hierarchical classification, we showed that this joint training is beneficial compared to the staged strategy of first positioning the prototypes and then training a feature extracting network. A PyTorch implementation of our framework as well as an illustrative notebook are available at `https://github.com/mgp-anon/metric-guided-prototype` (repository anonymized for review).

This work, along with other recent investigations (Bertinetto et al., 2020), highlights the interest of modeling hierarchical class structures in modern deep networks. Beyond the decrease in prediction costs, such hierarchies can lead to an improved overall performance. This calls for further investigation into the mechanisms by which semantic class structures can benefit the learning of expressive representations. In further research, we plan to investigate whether or claims hold in hyperbolic embedding spaces, which are known to be well-suited for embedding hierarchies.

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
