# OpenReview forum: "Leveraging Class Hierarchies with Metric-Guided Prototype Learning"
_ICLR.cc/2021/Conference — Reject_

### Official Review · AnonReviewer4 · 2020-10-13

**Rating:** 4
**Confidence:** 4

**Review:**

Summary:

This paper proposes a prototype-based approach to learning with prior hierarchical knowledge of categories. The main idea behind their approach is to perform metric-guided penalization on top of a prototype-based cross-entropy loss. The penalization is scale-free, yet the loss can be computed in closed form, allowing for a hierarchical guidance to prototype locations.

Strengths:

The starting motivation and direction of the paper are clear. If hierarchical knowledge is available, it is intuitive to exploit such information. Care has been taken to make the research open and reproducible. Along with the submission, an anonymized repo and notebook with details are provided. Both help with understanding the approach and its workings in practice.

The scale-free component of the proposed regularization is interesting, most notably the fact that the corresponding loss can be computed in closed form. By making the hierarchical loss scale free, a clash with the cross-entropy loss - which tries to push prototypes as far away as possible from each other - is avoided.

Weaknesses:

The main weakness of the paper is that its innovation is limited, especially in light of missed recent works on prototype-based approaches using hierarchies in non-Euclidean spaces. The innovation of the paper is in Section 3.2, which means that the technical novelty of the paper is limited to a loss with hierarchical loss. More pressingly, relevant recent literature is missed on using hierarchical knowledge in deep networks. What the authors consider future work ("Among other promising avenues for further research, we plan to investigate the use of non-Euclidean embedding spaces, which are known to be well-suited for embedding hierarchies [...]", page 8), has readily been proposed, see e.g.:

[1] Liu, Shaoteng, et al. "Hyperbolic Visual Embedding Learning for Zero-Shot Recognition." Proceedings of the IEEE/CVF Conference on Computer Vision and Pattern Recognition. 2020.
[2] Long, Teng, et al. "Searching for Actions on the Hyperbole." Proceedings of the IEEE/CVF Conference on Computer Vision and Pattern Recognition. 2020.
[3] Khrulkov, Valentin, et al. "Hyperbolic image embeddings." Proceedings of the IEEE/CVF Conference on Computer Vision and Pattern Recognition. 2020. (note: this paper does not explicitly utilize hierarchical knowledge)

All three above papers, from the same conference as the often cited work by Bertinetto et al. (2020) in the submission, investigate prototype-based approaches in non-Euclidean spaces. [1,2] explicitly use hierarchical knowledge to steer the prototypes. Hyperbolic spaces are known to be a better fit for hierarchies, see e.g. [4].

[4] Nickel, Maximillian, and Douwe Kiela. "Poincaré embeddings for learning hierarchical representations." Advances in neural information processing systems. 2017.

Compared to recent works on prototypes in hyperbolic spaces with hierarchical knowledge, the paper provides limited novelty. As the authors state that moving the prototypes to non-Euclidean spaces is future work for this submission, the other recent works make the paper come across as somewhat outdated.

Empirically, the paper provides a comparison to multiple hierarchical alternatives on four datasets. The use of four datasets is appreciated, as it provides a clear picture of the strengths and limitations of the approaches. Besides missing comparisons to e.g. the supervised setup of [2], Figures 2 and Table 2 show that the difference with alternatives is minimal. E.g. the guided prototypes seem to only outperform soft labels on iNaturalist-19 in terms of Average Cost. Table 2 paints a picture that fixing the scale or using a coarser ranking-based approach hardly affect the Average Cost on all datasets and only provide some improvements on one of the four datasets.

Conclusion:

The paper addresses an open problem in deep learning literature in a clear manner. The open science setup of the submission is appreciated and the scale-free regularizer is interesting. In light of recent advances in hyperbolic prototype-based approaches with hierarchical knowledge, the innovation of this work is minimal. The empirical benefits of the proposed approach are also not always convincing. For the rebuttal, it would be interesting to discuss and compare the submission to hyperbolic hierarchical prototype approaches.

Opinion post rebuttals:

After reading the rebuttal and the other reviews, I remain of the opinion that more work is needed before warranting acceptance. This paper indeed learns prototypes on the fly in contrast to e.g. Mettes et al. 2019. That method was however not designed for hierarchical knowledge, while several recent CVPR papers were. Since the novelty over these papers is limited and direct comparisons are lacking, more research is needed.

---

> ### Author Response · Authors · 2020-11-12
> **Reply to Reviewer 4**
>
> We want to thank you for the thorough review, and for the very interesting papers that you brought to our attention.
>
> While we compare ourselves to Mettes etal. 2019 which use an hyperspherical embedding, we seem to have missed the new hyperbolic methods released over the summer between CVPR2020 and ICLR2021. We will adjust thoroughly our claim and the related work section. We want to point out a key difference however between our work and [1], [2]: we learn simultaneously the prototype positions and the embedding network, while this is done in stages for [1], [2] as well as Mettes et al.2019. This staged-schedule (that we also reproduced in our setting in the fixed-proto method in the ablations) yields consistently inferior results for the Euclidean space. But we have yet to try it on hyperbolic spaces.
> The goal of our work is to show the advantage in learning simultaneously the feature extracting networks and an prototypes arrangement representing a metric derived from a hierarchy by adding a simple regularizer. Our first objective was to prove this thesis on Euclidean space, for which we reimplemented 5 recent competing methods from scratch.
> Lastly, we argue that our regularizer can be added at no parameter cost and with essentially no  hyper-parameters tuning. In this sense, it is conceptually and practically simpler than the more involved hyperbolic optimization, and avoids associated drawbacks such as numerical precision issues and multiple optimizer parametrization.
>
> Regarding the significance of the results, we invite you to read our response to Reviewer 1 (paragraph “Results significance”).
>
> *We modified our article according to yours and the other reviewers’ comments. In the second to last submitted version, the main parts that were changed are highlighted in red, and the most recent submitted version is the updated article.*

---

> > ### Comment · AnonReviewer4 · 2020-11-23
> > **Response**
> >
> > A difference with the missed CVPR 2020 papers is indeed that here, prototypes are learned as part of the network optimization. However, drawing direct comparisons is not possible without empirical evaluations. In this submission, the Euclidean space is used, while hyperbolic embeddings have in recent years already shown that the Euclidean space is suboptimal for hierarchies compared to hyperbolic spaces. The mentioned CVPR 2020 papers are a result of that. A difference in the training procedure does not discount that other works have readily done what you list as future work.
> >
> > I have also read the discussion on the significance of the results. The proposed approach does outperform the baselines on multiple settings, but the difference is small on aggregate. Also, the statement that reducing AC increases ER is a bit puzzling. If it is not expected that hierarchical knowledge can help reduce the error rate of the problem at hand (e.g. because of additional semantic disambiguation), the overall impact of hierarchical knowledge is limited.
> >
> > After reading the rebuttal and the other reviews, I remain of the opinion that more work is needed before warranting acceptance.

---

### Official Review · AnonReviewer2 · 2020-10-28
**Nice idea, providing consistent (small) accuracy improvements in well conducted experiments**

**Rating:** 6
**Confidence:** 3

**Review:**

Strengths and weaknesses:
+
Nice idea
Consistent improvements over cross entropy for hierarchical class structures
Improvements w.r.t other competitors (though not consistent)
Good ablation study
-
The improvements are small
The novelty is not very significant


More comments:


Figure 1:
-	It is not clear what distortion is at this stage
-	It is not clear what perturbed MNist is, and respectively: why is the error of a 3-layer CNN so high (12-16% error are reported)? CNNs with 2-3 layers can solve MNist with accuracy higher than 99.5%?
-	This figure cannot be presented on page 2 without proper definitions. It should be either presented on page 5, where the experiment is defined, or better explained
Page 4: It is said that s can be computed efficiently and this is shown in the appendix, but the version I have do not have an appendix
Page 6: the XE+EMD method is not present in a comprehensible manner. 1) p_k symbols are used without definition (tough I think I these are the network predictions p(\hat{y}=k|I)  2) the relation of the formula presented to the known EMD is not clear. The latter is a problem solved as linear programming or similar, and not a closed form formula  3) it is not clear what the role of \mu is and why can be set to 3 irrespective of the scale of metric D
page 7:
The experiments show small, but consistent improvements of the suggested method over standard cross entropy, and improvements versus most competitors in most cases

I have read the reviews of others and the author's response. My main impression of the work remains as it was: that it is  nice idea with small but significant empirical success. However, my acquaintance with the previous literature in this subject is partial compared to the acquaintance of other reviewers, so It may well be possible that they are in a better position than me to see the incremental nature of the proposed work. I therefore reduce the rating a bit, to become closer to the consensus.

---

> ### Author Response · Authors · 2020-11-12
> **Reply to Reviewer 2**
>
> Thank you very much for your review and for your positive feedback.
> - **Figure 1** We agree that the caption of Figure 1 is not as clear as it should, we will try to improve it in our revised version.
> - **Appendix** the appendix is in the supplementary material zip files.
> - **Performance on MNIST** the performance is lower than SOTA networks because we restricted ourselves to a feature map of dimension 2  for visualization purposes. Furthermore, we added random flip and affine transform as augmentation.
> - **EMD** Your interpretation of p_k is correct, we will fix it in the revised paper. In the case of this method the EMD is computed between a class distribution and a one-hot-encoding which can be computed in close-form. \mu is a parameter described in Hou etal 2016, and that we have fine-tuned along the regularization weight.
>
> *We modified our article according to yours and the other reviewers’ comments. In the second to last submitted version, the main parts that were changed are highlighted in red, and the most recent submitted version is the updated article.*

---

### Official Review · AnonReviewer3 · 2020-10-28
**Reinvention of several known concepts; small scientific contribution**

**Rating:** 4
**Confidence:** 5

**Review:**

In order to organize my review, I use the NeurIPS2020 template with slight adaptions to fit the ICLR requirements.

### Summary and contributions: Briefly summarize the paper and its contributions.

The paper proposes a method to jointly learn a feature-extractor (a neural network) and a set of prototypes. Learning is performed in an end-to-end setting and the prototype distribution is regularized to follow a given distribution. For this purpose, the authors propose a cost matrix that assesses the severity of errors between classes. This matrix is organized in such a way that it yields a finite metric. The distribution of prototypes is regularized such that the distances between prototypes reflect this finite metric. In several experiments, the authors evaluate the advantages of such a setting with respect to an average cost measure.

### Strengths: Describe the strengths of the work.

The work is well motivated and how the entire network setup is realized is easy to understand and straightforward.


### Weaknesses: Explain the limitations of this work along the same axes as above.

A major weakness is that the authors have not cited all the relevant previous work about prototypical networks (see my comments below). Taking all the relevant literature into account, the contribution of the paper is small. Moreover, the proposed method and observed effects raise several questions that are not sufficiently addressed. For example, the authors observe that the learning with the squared Euclidean distance does not perform as good as with the Euclidean distance, but they have not investigated why this is the case (see my further comments below).


### Correctness: Are the claims and method correct? Is the empirical methodology correct?

The current version of the paper has some minor mathematical mistakes. However, they can be fixed and I consider them to be of little relevance for the correctness of the method. I will list them below.
One serious correctness point is that the authors claim (see page 4 top and page 2 contributions first bullet point) that they are inventing a principle to learn the prototypes jointly in an end-to-end setting. Considering the state of the art, this claim is incorrect.


### Clarity: Is the paper well written?

The paper is well written and the mathematical formulations have a good quality.


### Relation to prior work: Is it clearly discussed how this work differs from previous contributions?

The paper has not discussed all the relevant prior work. The idea to learn the prototypes jointly is not new:
* Chen, Chaofan and Li, Oscar and Tao, Daniel and Barnett, Alina and Su, Jonathan and Rudin, Cynthia. “This Looks Like That: Deep Learning for Interpretable Image Recognition.” NeurIPS (2019).
* Hong-Ming Yang and Xu-Yao Zhang and Fei Yin and Cheng-Lin Liu. “Robust Classification with Convolutional Prototype Learning.” CVPR (2018)
* Villmann, Thomas and Biehl, Michael and Villmann, Andrea and Saralajew, Sascha. “Fusion of deep learning architectures, multilayer feedforward networks and Learning Vector Quantizers for deep classification learning.” WSOM+ (2017)
* to name just a few…

Note that prototype-based classifiers and especially nearest prototype classifiers have been invented long before Tibshirani et al. (2002). For example, see Kohonen’s Learning Vector Quantization (LVQ) algorithms, including the original heuristic versions and the end-to-end trainable versions like Generalized LVQ (by Sato and Yamada).
Moreover, the incorporation of a cost matrix into the learning processes for prototypes was also done before. For example, Kaden, Marika, Wieland Hermann, and Thomas Villmann. "Attention Based Classification Learning in GLVQ and Asymmetric Misclassification Assessment." Advances in Self-Organizing Maps and Learning Vector Quantization. Springer, Cham, 2014. 77-87. Of course, in this work, the authors have not used the cost matrix to penalize the prototype positions, but it was included to account for different severity between class confusions.

### Reproducibility: Are there enough details to reproduce the major results of this work?

The paper provides a lot of details about the experimental setup, especially in the appendix. However, some information is not provided, for example, the initialization strategy of the architecture.


### Additional feedback, comments, suggestions for improvement, questions for the authors, and a **recommendation (accept or reject)**.

The authors should be precise about the assumptions of $D$. In the current version, the definition of $D$ is not correct to talk about a finite metric space (see page 3 Method and page 1). The missing property is the identity of indiscernibles.

The authors provide a good motivation for why the incorporation of a cost matrix is useful. However, could the authors explain why they then assume that the matrix $D$ is symmetric? Because, according to the motivation, the severity of misclassifying a *crossing pedestrian* as a *street lamp* is different from misclassifying a *street lamp* as a *crossing pedestrian*.

The first sentence in Section 3.1 is difficult to read due to “sample $n$”, consider writing $x_n$.

What is the difference of Equation (3) to a cross-entropy loss with a one-hot encoding for the correct class?

The authors observed that the squared Euclidean distance does not perform as well as the Euclidean distance. I assume that this happens because the squared Euclidean distance does not fulfill the triangle inequality that is, on the other hand, assumed for $D$. Could the authors comment on this?

I don’t see that the non-differentiability of the square root at zero causes any issues in the training. If the argument becomes zero, the distance is zero and, therefore, the prototype is equal to the feature vector of the sample. Given the given interpretation of the attraction and repulsion forces on the data and the prototypes (see page 3 below), in which direction should they be pushed or repelled if they have a perfect match? There exists no update direction that could improve the model. Therefore, it is feasible to just stop the gradient or to ignore such samples.

Have the authors observed training instabilities due to large values of the distance $d$?

If I consider Figure 1, I wonder why the learned prototypes tend to be outside the class centers (e.g., see figure c). Can the authors elaborate on that?

The statement on page 1 bottom about the cross-entropy is incorrect. Whether the cross entropy “singles out the prediction” depends on the true probability vector.

The experimental results are not convincing as they don’t show a significant benefit compared with state-of-the-art methods.

I have problems with the statement following Equation (4). Could the authors explain in more detail why a prototype arrangement regarding $D$ and low distortion is advantageous in terms of AC?

The google colab link provided in the supplementary was not accessible.

Considering all the above points, I vote for rejection, since the paper does not have the right maturity level to be published at ICLR and the scientific contribution is small.

---

> ### Author Response · Authors · 2020-11-12
> **Reply to Reviewer 3**
>
> Thank you for this thorough review, and for pointing out crucial papers that we have unfortunately missed.
> - **Missing bibliography**
> We recognize that the joint learning of prototypes has been already presented, we will remove all claims on that matter, and give proper credit.  We will hence refocus this work’s messaging on the metric-guided scheme we propose to incorporate class hierarchies into a prototype arrangement. Thank you also for pointing out interesting papers on the incorporation of error cost.
> - **Reproducibility** all details not specified are default PyTorch options. Note that the code is available in an anonymized repository: https://github.com/mgp-anon/metric-guided-prototypes
> - **Structure of D** Thank you for pointing to the missing property of D, we will correct this sentence. Regarding the symmetry of the cost matrix, we invite you to read our response to Rev1 who also raised this point.
> - **Prototypes arrangement** as remarked in 4.4, the class of sample embeddings follow a Voronoi partition with respect to the prototypes rather than being organized evenly around a centroid prototype. Recall that samples are classified according to the softmax of the distance between their representations and all prototypes.  Consequently, only the relative difference between these distances determine the posterior, rather than the distance to the closest prototype alone. A very confident sample will not try to exactly coincide with its prototype, but also to maximize the distance with other prototypes. Therefore, any direction which preserves the delta of distance will produce the same posterior. This is why samples tend to be 'behind'  rather than 'around' their class prototype. We argue that this particularity is mostly overlooked by mean-based prototypes.
> - **Effect of low distortion** We argue that prototypes arrangements \pi with low distortion incur lower hierarchical costs. Let us first consider a misclassified sample x_n of true class k. Since x_n is misclassified, its representation f(x_n) is closer to another prototype than to the true prototype \pi_k. However, we can assume that f(x_n) is still closer to \pi_k than to most prototypes, as the loss L_data would starkly penalize f otherwise. Since low distortion prototypes group together prototypes of classes with small mutual error cost, the erroneous class predicted will likely be of low hierarchical cost with respect to k.
> - **Equation 3** is equivalent to the distance-based cross entropy. It does not involve a linear mapping between a feature map and a vector of class scores, but the relative position of a representation and a prototype.
> - **L2 distance** we observed indeed that training prototypical networks with L2 distance lead to less stable training. We attribute this phenomenon to the tendency of the L2 norm to focus too much on large distances. This is especially relevant in our setting, for which the local arrangement of prototypes has the most influence, and confusions with distant prototypes are rare by design
> - **Performance** we refer the reviewers to our answer to Rev1 (paragraph “Results significance”).
> - **Huber norm** our idea was to avoid instability for distances very close to 0 in which the gradient of the norm “overshoot” the desired equality, resulting in a cyclical update loop. However, the effects of the Huber norm were minimal.
>
> *We modified our article according to yours and the other reviewers’ comments. In the second to last submitted version, the main parts that were changed are highlighted in red, and the most recent submitted version is the updated article.*

---

> > ### Comment · AnonReviewer3 · 2020-11-23
> > **Response**
> >
> > Thanks to the authors for the response.
> >
> > From my point of view, the paper adds a minor scientific contribution that is not carefully enough evaluated (see also the comments by R4 and R1). Therefore, I keep the score of my initial evaluation (also because of the good comments by R1 and R4).
> >
> > Moreover, please note the following comments:
> > * Equation 3: It is just a cross-entropy as the output of the proposed architecture is defined as the vector of distances to the  prototypes.
> > * L2 distance: If the authors observed such things, it should be mentioned in the paper. Additionally, please discuss these results according to the issues known for the L2 distance (e.g., Ghiasi-Shirazi, Kamaledin "Generalizing the Convolution Operator in Convolutional Neural Networks" Neural Processing Letters 2019 --> scaling matters!).
> > * Prototypes arrangement + Effect of low distortion: "We argue that prototypes arrangements \pi with low distortion incur lower hierarchical costs." I doubt that this statement is true. I think the main benefit comes from fixing the scale of the distances such that the prototype distribution cannot explode (note the issue described by Xing et al (2003) "Distance metric learning with application to clustering with side-information"). Could the authors elaborate on that? Please also note the publications about "boarder sensitivity" in prototype-based methods (Kaden et al. (2015) "Border-sensitive learning in generalized learning vector quantization: an alternative to support vector machines"), which is similar to the described effect why the prototypes aren't like means.
> > * Huber norm: Have the authors observed this effect in practice and can provide an example? From my personal experiences, I never had issues with that.

---

### Official Review · AnonReviewer1 · 2020-10-28
**assumptions are not adequately discussed and results are inconclusive**

**Rating:** 4
**Confidence:** 3

**Review:**

The paper proposes a method to integrate the cost of errors into a classification algorithm. The proposed loss function has two terms: one for pushing samples towards their corresponding class prototype (L_{data}), and another for forcing/guiding the pairwise distance between class prototypes to follow a predefined values (L_{distortion}).

Marginally below acceptance.

I like the idea of using a priory knowledge about similarities between classes to regularize the learning process.  But,

1) Out of 4 datasets, the proposed method works better than others on only two datasets for which in one case the addition of proposed term L_{distortion} has helped, but it actually degraded the results in the other dataset. So, the results are inconclusive.

2) pp.3 the assumption of having a symmetric misclassification cost seems very limiting. Effectively, this assumption implies the type 1 and type 2 errors must have equal cost. The proposed method is about treating different errors differently and yet it assumes type 1 and type 2 errors are equal. Referring to the example, provided in the introduction (paragraph 1), cost of misclassifying a crossing pedestrian as a street light should be way more than the cost of classifying a street light as a crossing pedestrian.
Another assumption of the proposed method is triangle inequality i.e. D[k, l] + D[l,m] >= D[k,m] but this assumption is not justified except saying the cases where the assumptions do not hold, are out of scope of this paper.

pp.1 says, "this error discrepancy is not taken into account ... in the evaluation metrics". This statement is incorrect. At least in a simple binary classification task, we have type 1 and type 2 errors. We have recall and precision. They are all about putting different weights to different errors.

pp.2 Caption of Figure 1: tree is created based on authors perceived visual similarity. The tree should reflect the cost of class confusions. From paragraph 1 in page 1, it is inferred that the cost of a class confusion is determined by the actual consequences e.g. cost of confusing a streetlamp with a crossing pedestrian. Such cost is different than how similar the two classes are.  The notion of "cost" and "similarity" are not the same but used interchangeably in the paper.

pp.7 the soft-label method is the top runner in cifar-100. Its performance on the other 3 datasets may be improved if the temperature parameter is adjusted to compensate the various number of classes. Of course, same can be said about tuning hyperparameters of the proposed method.

---

> ### Author Response · Authors · 2020-11-12
> **Reply to Reviewer 1**
>
> We thank you for your in-depth and careful review, which will help us to present more clearly the focus of our method.
> - **Cost/Similarity and symmetry of the cost matrix**
> In all our numerical experiments the error cost is derived from a distance in a tree-shaped hierarchy of classes, which implies the symmetry of the cost. While our formalism technically allows us to deal with the more general setting of a cost matrix derived from a finite metric, we agree that we haven’t sufficiently argued why this setting would be useful, and our autonomous driving exemple is misguided. We reframe the motivation accordingly by focusing on cost matrices derived from hierarchies.
>
> - **Results significance**
> We want to insist that the focus of our paper is to reduce the hierarchical cost AC. It is expected that focusing on AC should increase the error rate ER. In that regard, our method performs best in 3 out of 4 datasets while being a very close second in the last. Moreover, our method is the only one which outperforms the traditional cross-entropy setting for all datasets and metrics.
> Furthermore, our regularization can be added to prototypes to improve the AC with no extra parameters and close to no computational cost. The automatic scaling allows us to use the same hyper-parameterization across 4 vastly different datasets (superpectral time series vs RBG image), tasks (picture to pixel to parcel classification), and hierarchies (with depth varying from 3 to 7) . On this matter, the hyperparameters used for the soft-label method is one of the recommended values in Bertinetto et al. for iNat-19.
> - **Recall**  We agree that the precision and recall measure with different types of error. However, we argue that (in the multiclass setting), this measure doesn’t take into account class specificity and treats all type I or II errors the same way.
>
> - **Cost in MNIST** In Figure 3 of the appendix we present a variation of the MNIST experiment in which the cost matrix is defined as the absolute difference between digit values as you suggest. This experiment shows that when the cost matrix does not contain semantic insights, prototype guiding is detrimental to the overall accuracy. However, the AC does improve with our method.
>
> *We modified our article according to yours and the other reviewers’ comments. In the second to last submitted version, the main parts that were changed are highlighted in red, and the most recent submitted version is the updated article.*

---

> > ### Comment · AnonReviewer1 · 2020-11-23
> > **inconclusive results**
> >
> > Re Results significance: The authors say "It is expected that focusing on AC should increase the error rate ER". Focusing on average hierarchical cost means additional information (the class hierarchy) is used. In other words, the authors are saying they use additional information and yet the performance degrades, and this normal?! I don't find this response acceptable for justifying the inconclusive results reported in the paper (see item 1 in my original response).

---

> > > ### Author Response · Authors · 2020-11-23
> > > **Clarification**
> > >
> > > The cross-entropy loss with one hot target vector is a direct surrogate of the Error Rate metric in the sense that both treat all class confusions equally. Altering this evenness may result in a worse performance with regard to the ER, similarly to class balancing schemes. Our method's objective is primarily to reduce the AHC by informing the networks that some confusion costs more wrt this metric.
> > > When the hierarchical structure contains valuable information about the nature of classes, this can lead the model to learn better features, and ultimately counterbalance the decrease in ER that would be caused by altering the cross-entropy.
> > > In short, our method aims at improving the AHC, which we argue has value in itself. Occasional improvement in ER are a fortuitous consequence of well-chosen hierarchies, and not benefit to expect systematically.

---

### Decision · Program_Chairs · 2021-01-07
**Final Decision**

**Decision:**

Reject

**Comment:**

Thank you for your submission to ICLR.

This paper had somewhat dissenting reviews, but three of four reviewers felt negatively about the paper.  On the positive side, the reviewers noted good motivation for the problem, a good ablation study and, in some cases, good performance over standard cross entropy.  On the negative side, the reviewers noted limited novelty, missing discussion or comparison to prior work, and in some cases only marginal improvement over existing methods.

The positive reviewer still remained positive after discussion, but noted that their confidence was low on the paper and that they would defer to others' opinions.  The other reviewers still had several concerns after the discussion phase.  Ultimately, it seems that the paper could use some additional work before it is ready for publication.  I would strongly encourage the authors to keep the reviewer comments in mind when preparing a future version of the manuscript.